# Usual Nutrient Intake Distribution and Prevalence of Inadequacy among Australian Children 0–24 Months: Findings from the Australian Feeding Infants and Toddlers Study (OzFITS) 2021

**DOI:** 10.3390/nu14071381

**Published:** 2022-03-25

**Authors:** Najma A. Moumin, Merryn J. Netting, Rebecca K. Golley, Chelsea E. Mauch, Maria Makrides, Tim J. Green

**Affiliations:** 1Discipline of Pediatrics, Faculty of Health and Medical Sciences, University of Adelaide, Adelaide, SA 5000, Australia; najma.moumin@sahmri.com (N.A.M.); merryn.netting@sahmri.com (M.J.N.); maria.makrides@sahmri.com (M.M.); 2Women and Kids Theme, South Australian Health and Medical Research Institute, Adelaide, SA 5000, Australia; 3Nutrition Department, Women’s and Children’s Health Network, Adelaide, SA 5006, Australia; 4Caring Futures Institute, College of Nursing and Health Sciences, Flinders University, Adelaide, SA 5000, Australia; rebecca.golley@flinders.edu.au (R.K.G.); chelsea.mauch@flinders.edu.au (C.E.M.)

**Keywords:** Australian feeding infants and toddlers study, infants, toddlers, nutrient intake, nutrient reference values, prevalence of inadequacy, Australia, survey

## Abstract

(1) Background: Breastmilk provides all the nutrition an infant requires between 0–6 months. After that, complementary foods are needed to meet the child’s increasing energy and nutrient requirements. Inadequate energy and nutrient intake may lead to growth faltering, impaired neurodevelopment, and increased disease risk. While the importance of early life nutrition is well recognized, there are few investigations assessing the nutritional adequacy of Australian children <24 months. Here, we describe usual energy and nutrient intake distributions, including the prevalence of inadequate intakes and exceeding the upper limit (UL), in a national sample of Australian children 6– 24 months and infants < six months who had commenced solids and/or formula. (2) Methods: Dietary intakes were assessed using a one-day food record for 976 children with a repeat one-day record in a random subset. (3) Results: Based on the Nutrient Reference Values for Australia and New Zealand, children’s intakes were above the Adequate Intake or Estimated Average Requirement for most nutrients. Exceptions were iron and zinc where the prevalence of inadequacy was estimated to be 75% and 20%, respectively, for infants aged 6–11.9 months. Low iron intake was also observed in one quarter of toddlers 12–24 months. On average, children consumed 10% more energy than predicted based on Estimated Energy Requirements, and ~10% were classified as overweight based on their weight for length. One third of toddlers exceeded the tolerable upper limit for sodium and consumed > 1000 mg/day. Of the children under six months, 18% and 43% exceeded the UL for vitamin A (retinol) and zinc. (4) Conclusions: Compared to nutrient reference values, diets were sufficient for most nutrients; however, iron was a limiting nutrient for infants aged 6–11.9 months and toddlers 12–24 months potentially putting them at risk for iron deficiency. Excessive sodium intake among toddlers is a concern as this may increase the risk for hypertension.

## 1. Introduction

The first two years of life are a period of rapid growth and development. For the first six months of life, breastmilk is sufficient to meet nutritional requirements; however, at around six months, complementary (solid) foods are needed to meet increased energy and nutrient needs for growth [1,2,3]. On a per kg basis, infants require three times as much energy and five times as much iron, zinc, and vitamin A as an adult [4]. Failure to receive sufficient energy and nutrients from complementary foods may lead to growth faltering, cognitive impairment, and increased risk of infection [5,6]. Conversely, excessive energy intake may lead to rapid weight gain and excess energy from protein has been linked to obesity in later life [3,7,8].

In Australia, children less than two years old have been excluded from national nutrition surveys [9,10,11]; therefore, little is known about their diets. A small number of Australian studies have reported on energy and nutrient intakes in this age group [12,13,14,15,16], and iron and zinc have been identified as problem nutrients. However, the data for these studies were collected 10–15 years ago and were often limited to one or two Australian cities. Therefore, contemporary data on population intakes are urgently needed to better understand the scope of the problem and to identify evidence-based solutions to improve dietary intakes. To this end, we aim to describe mean energy and nutrient intakes of the population, including the prevalence of inadequate intakes based on Nutrient Reference Values (NRVs) for Australia and New Zealand [4] in a national sample of Australian children 0–24 months. Because breastmilk is assumed to meet infants’ nutritional needs for the first six months, energy and nutrient intakes were only assessed for those infants receiving breast milk substitutes (formula) and/or consuming complementary foods.

## 2. Materials and Methods

A detailed description of the OzFITS 2021 data collection procedures and survey methods can be found elsewhere [17]. OzFITS 2021 was an Australia-wide cross-sectional survey conducted between April 2020 and April 2021. In total, 1140 caregiver-child dyads were enrolled over the 12 months. The study was approved by the Women’s and Children’s Health Network Human Research Ethics Committee (HREC/19/WCHN/44), and all caregivers gave informed verbal consent.

Caregivers with eligible children were referred to the study via a trial recruitment company [18]. At enrolment, all caregivers completed a sociodemographic and child-feeding questionnaire adapted from the 2016 United States Feeding Infants and Toddlers (FITS) and the 2010 Australian National Infant Feeding Survey [19,20]. Caregivers with children 0–24 months who had commenced complementary foods or consumed formula (*n* = 976) were invited to complete a food record for their child. Since breastmilk is sufficient to meet energy and nutrient needs for infants less than six months, caregivers with infants receiving only breastmilk at enrolment (*n* = 164) were not asked to complete a food record. Instead, a detailed breastfeeding history including the average number of breastfeeds in a 24-h period was collected. All survey data were collected and managed using REDCap™ (Research Electronic Data Capture) [21,22].

Dietary intake was estimated using a one-day food record, with a random subset of the population (~30%) completing a second food record on a non-consecutive day. Caregivers were randomly assigned to one or two record keeping days and then sent a study package, by post, which included a food record booklet and portion size estimation guide [10,19]. Once the study package was delivered, caregivers received a preparatory phone call from staff where they were instructed on how to record the portion size of foods and drinks offered using standard metric cup and spoon measures, kitchen scales, and the portion estimation guide. They were then asked to record everything their child consumed in a 24-h period starting from midnight using the food record booklets provided. Once completed, caregivers were asked to take photos of the booklet(s) and scan or email them to study staff. Interviewers then contacted the caregiver to review the food record(s) for completeness and accuracy. Interview techniques described in the four-pass 24-h recall method were used to systematically review food records with caregivers [23]. Pass one involved reviewing all the foods and drinks listed on the food record, followed by a detailed description of each item i.e., preparation method, brand name, variety etc. Portion sizes consumed, inclusive of leftover amounts and spills, were then confirmed in pass three. Finally, in pass four, the food record was reviewed in full including probing for forgotten foods and snacks.

All food intake data was entered into FoodWorks™ Professional Version 10, a dietary analysis software program [24] which uses the 2011–2013 Australian Food, Supplement and Nutrient Database [25]. Commercial infant and toddler foods that were not available in FoodWorks™ were calculated using nutrient information and ingredient lists from the nutrition information panel. Most micronutrient values were not included on the nutrition information panel; therefore, a recipe approach based on ingredient lists was used to estimate these values [26]. Briefly, ingredients were entered as cooked food items in FoodWorks™ based on their proportion within the ingredient list (product recipe). Different quantities were imputed for individual ingredients until a nutrient profile that closely matched the nutrition information panel was achieved. All product recipes imputed in this way were within 10% of the manufacturer’s reported energy, total fat, carbohydrate, protein, and total sugars. For fortified products, nutrient values generated by FoodWorks™ were replaced with the manufacturer’s reported values.

Breastmilk intakes were estimated based on validated assumptions used in previous studies [2,27,28]. The number of minutes of active feeding were recorded for each breastfeed and converted to a fluid volume of 12.5 mL/min for infants < 6 months and 10 mL/min for older infants and toddlers up to a maximum of ten min per feed [29]. Feeds less than two minutes were excluded. Energy and nutrient content/100 g breastmilk was then calculated and entered as a food item in FoodWorks™ based on human milk composition data [4]. Expressed breastmilk was entered as the quantity expressed and fed to the child. For infant formula and toddler milk, the amount consumed was calculated from the caregiver’s reported method of preparation using the following formula: formula (g) = scoop weight (g)/prepared volume (mL) × consumed volume (mL).

Caregivers were asked to provide their child’s current length (cm) and weight (kg). If the child’s length and weight had been measured by a healthcare professional in the previous 30 days, these measures were used. Otherwise, caregivers were asked to measure length and weight, according to instructions provided for in-home measurements [30].

### Data Analysis

Quality assurance procedures ensured data entry accuracy and investigators completed a line-by-line audit of 10% of food records. Under or overreporting of energy intakes was estimated by comparing energy intakes to age-specific estimated energy requirements (EER) [4]. Energy intakes (EI) were considered plausible if EI: EER was between 0.54–1.46 [15,31]. Two investigators reviewed food records and case notes for all extreme under or over reporters and only two were removed from analysis.

Z-scores for weight for length were calculated using the World Health Organization Anthro Survey Analyzer™ [32], which uses child growth standards developed from the WHO Multicenter Growth Reference Study [33]. Children who were ≥+1 SD, ≥+2 SD, and ≥+3SD above the mean were classified as at risk for overweight, overweight, or obese, respectively [33]. Prevalence estimates with 95% CI were calculated.

Within-person variability in nutrient intake was adjusted using the Iowa State University method with the Intake Modelling, Assessment, and Planning Program (IMAPP) software [34]. Usual intake distributions were compared to the NRVs for Australia and New Zealand. For infants < 12 m, most nutrients lack an estimated average requirement (EAR); therefore, the probability of meeting adequate intakes (AI) is reported [4]. To estimate the prevalence of inadequate and excessive nutrient intakes, the EAR cut points, and tolerable upper limits (UL) described in the NRVs for Australia and New Zealand [4] were used. Since the distribution for iron is asymmetrical about the EAR, the cut point method cannot be used [35]. Thus, to estimate the risk of inadequacy for iron, the ‘full probability approach’ was used and compared the distribution for usual iron intake to the iron requirement distribution percentiles at 10% and 15% bioavailability for infants 6–11.9 months and toddlers 12–24 months, respectively [4,35]. Although the NRVs estimate 14% iron absorption for toddlers based on dietary modelling, this absorption percentile was not available in IMAPP, therefore the closest available percentile (15%) was used. Nutrient intake from supplements was not included, as only 6% percent of caregivers reported supplement use.

## 3. Results

Characteristics of the infants and toddlers eligible for food records are described in Table 1. Between April 2020 and April 2021, 976 caregivers with children < 24 months were enrolled and asked to keep a food record for their child; 29 withdrew and 94 were lost to follow before completing the food record. 853/976 (87%) completed the first food record, and 290/345 (84%) completed the second food record, which enabled the estimation of usual nutrient intake distribution. Breastfeeding rates were high in both infant age groups, and 210/542 (44%) of toddlers were still receiving breastmilk on the day of the food record. Nearly 40% of all children were classified as at risk for being overweight, with 10% classified as overweight. Underreporting of energy intake, an EI: EER below 0.54, was only observed in two children. In contrast, 58/851 (7%) of children had EI: EER above 1.46, suggesting a high energy intake. On average, energy intakes were 10% higher than EER for all age groups.

Table 2 describes the mean nutrient intakes and prevalence of inadequacy or excessive intake for infants 0–5.9 months consuming formula and/or complementary foods. Two infants had energy intakes more than twice the EER and were excluded from analysis. Average daily energy intakes were approximately 2553 ± 60 kJ/day, with carbohydrates and fat contributing equally to energy intake. For most nutrients a high proportion of infants came close to meeting the AI, except for iodine (69%). The UL for retinol and zinc was exceeded by 18% and 43% of infants, respectively.

The usual energy and nutrient intake distributions for infants aged 6–11.9 months and toddlers 12–24 months are described in Table 3 and Table 4. More than 75% of older infants and one quarter of toddlers had inadequate iron intakes (Table 3 and Table 4). Approximately one-fifth of older infants were estimated to be at risk for dietary zinc inadequacy. One third of toddlers exceeded the UL for sodium.

## 4. Discussion

This is the first national study to estimate usual energy and nutrient intake distribution, including the prevalence of inadequate and excessive intakes for key micronutrients, in Australian children under two years of age. Overall, children met or exceeded AIs or EARs and were below the UL for most nutrients. Exceptions were a very high prevalence of inadequacy for iron (75%) in infants 6–11.9 months and excessive sodium consumption in a high proportion of toddlers 12–24 m (30%).

Because breastmilk is sufficient to meet the nutritional needs of infants < 6 months, we only assessed dietary intakes of infants receiving breastmilk substitutes and/or solid foods. For these infants, a high proportion met their AI, ranging from 69% for iodine to 100% for niacin. Intakes above the AI are considered adequate to meet nutrient requirements; however, intakes below the AI do not indicate nutritional inadequacy as they are based on intakes of apparently healthy groups or populations rather than an EAR [4,36]. Consistent with US FITS 2016 [36], a high proportion of infants in our study exceeded the UL for vitamin A (18%) and zinc (43%) which can be explained by the higher concentrations of these nutrients found in breastmilk substitutes compared to breastmilk. However, given the lack of reports of widespread hypervitaminosis A and zinc toxicity in Australian and US infants, this does not appear to be a problem and the ULs may be set too low [37].

For nutrients with AIs, the proportion of infants (6–11.9 months) meeting or exceeding requirements ranged between 33% for iodine to 100% for niacin. For the two nutrients with EARs, iron and zinc, the prevalence of inadequacy was estimated to be 75% and 20%, respectively. Compared to other life stages, the EARs for iron (7 mg/d) and zinc (2.5 mg/d) are exceptionally high, and inadequate intakes are frequently reported for this age group in other high-income countries [38,39,40]. Nevertheless, compared to our survey, previous Australian and American studies report much lower prevalence of inadequacy for both iron (9–36%) and zinc (1–9%) [12,15,36]. The reasons for this may be attributable to differences in infant formula and breastmilk consumption. In our study, >75% of infants were breastfed whereas 55–70% of infants in US FITS 2016 [36] and other Australian studies [12,15] consumed infant formula as their primary milk source. Because infant formula is fortified, breastfed infants are more reliant on nutrient dense foods to meet their iron and zinc requirements and are more likely to have inadequate intakes [2,3]. However, whether the high prevalence of dietary inadequacy is reflected in biomarkers of iron status or anemia is unknown. There are no contemporary data on the prevalence of iron or zinc deficiency in Australian infants aged 6–12 months. The only reports are from the 1990s and are from small samples in Sydney and Adelaide, which only assessed iron status [41,42,43].

EARs have been set for most nutrients for toddlers (one to two years). Apart from iron, the prevalence of inadequacy for all nutrients was low, around 3%. The prevalence of iron inadequacy in this age group is 25%, which is much lower than for infants and is more consistent with other Australian studies [12,13,14,16] and the US FITS 2016 [36]. For example, Zhou et al. reported a similar prevalence of inadequate iron intakes (16%) in a representative sample of Adelaide toddlers [13]. The decrease in the prevalence of inadequacy for iron from late infancy to toddlerhood can be attributed to the lower EAR for this age group (4 mg/d vs. 7 mg/d), increased food consumption, and in some cases, consumption of fortified milks. Like infants, there are no national prevalence estimates for iron deficiency or iron deficiency anemia in Australian toddlers. However, the reported prevalence of iron deficiency or iron deficiency anemia (13%) closely matched the estimated prevalence of inadequate iron intake in the Adelaide study [13]. Of concern is the high proportion of toddlers (30%) exceeding the upper limit for sodium consumption. Although the evidence base is equivocal, exposing young children to excess salt may increase their preference for salty foods later in life [44]. A high salt intake is a known risk factor for hypertension and cardiovascular disease in later life [4].

A key strength of this study was the direct data capture afforded by the food record. Although the 24-h recall method has the advantage of the element of surprise, food records reduce recall bias and allow for more accurate estimations of portion size due to real time data capture [45,46]. In our study there was no evidence of under reporting, which is a common problem in dietary assessment. If anything, there was an over estimation of energy intake compared with estimated requirements by 10% in all age groups. This is also reported in the US FITS 2016 [36]. We did not obtain dietary intake data from infants < six months receiving only breastmilk, and those with complete diet records were slightly older, formula fed, and more likely to have commenced complementary feeding compared to breastfed infants. For this reason, results for infants <six months should be interpreted with caution.

We caution that the NRVs for Australia and New Zealand are based on a limited evidence base. Most were adopted from the United States Institute of Medicine, Dietary Reference Values that were established more than 20 years ago [35]. Most EARs and AIs are either based on breastmilk nutrient composition from a small number of mothers or extrapolated from other age groups [4,35].

A challenge inherent in assessing dietary intake in this age group is obtaining accurate estimates of breastmilk intake. Like other studies, we have relied on published assumptions to estimate the volume of breast milk consumed. These assumptions do not take into account variation in breastfeeding efficiency between infants and maternal differences in breastmilk production [29]. Moreover, breastmilk composition is variable, and this variation is not reflected in the nutrient composition tables [4].

We did not assess the dietary intake of Vitamin D in our study due to a lack of food composition data. We recognize that the intake of vitamin D in breastfed infants is likely inadequate due to the low content in breastmilk. Unlike other high-income countries, Australian health authorities do not recommend routine vitamin D supplementation of breastfed infants. We did not include the contribution of infant vitamin and mineral supplements in our nutrient estimates; however, less than 10% of children were given supplements, mainly Vitamin D. We also did not assess maternal vitamin and mineral supplement use. This may have led to an underestimation of the proportion of infants 6–11.9 m meeting the AI for iodine. In Australia, lactating women are advised to take iodine supplements to increase their breastmilk iodine concentration [47].

We acknowledge that our sample is not representative of the Australian population. Our population is more educated and economically advantaged. However, our breastfeeding rates and durations as well as use of breastmilk substitutes is consistent with the 2010 Australian National Infant Feeding survey [20].

## 5. Conclusions

In sum, children’s diets were adequate for most nutrients except iron, zinc, and sodium. Of concern is the very high prevalence of inadequate iron intake amongst infants. We urgently need a nationally representative dietary survey including nutritional biomarkers for iron and other key nutrients in young children. Diet quality, including food sources of limiting nutrients, will be addressed in this publication series. Finally, a global effort towards establishing more robust dietary reference values for infants and children is required.

## Figures and Tables

**Table 1 nutrients-14-01381-t001:** Characteristics of children enrolled in OzFITS 2021 eligible for food records (*n =* 976) ^1^.

Indicators	0–5.9 Months (*n* = 126) ^2^	6–11.9 Months (*n* = 308)	12–23.9 Months (*n* = 542)
Age (months), mean ± SD	4.4 ± 1.3	8.5 ± 1.7	17.7 ± 3.3
Sex (female), *n* (%)	58 (46.0)	152 (49.4)	252 (46.5)
Weight status % (95% CI) ^3^			
Risk of overweight	24.3 (15.9, 32.8)	41.2 (35.1, 47.4)	38.6 (33.9, 43.3)
Overweight	9 (3.2, 14.8)	9.9 (6.1, 13.7)	12.8 (9.5, 16.1)
Obese	4.5 (0.2, 8.8)	1.5 (0, 3.2)	2.3 (0.8, 3.9)
Eligible to complete a food record, *n* (%)	126 (43.4)	308 (100)	542 (100)
Asked to complete a 2nd day	47 (37.3)	110 (35.7)	188 (34.7)
Completed food record, *n* (%) ^4^			
1st day	114 (90.5)	279 (90.6)	460 (84.9)
2nd day	46 (97.9)	98 (89.1)	146 (77.8)
Milk feeding type ^5^			
Breastmilk	65 (70.7)	221 (77.3)	210 (44.2)
Infant formula	76 (82.6)	39 (13.6)	14 (2.9)
Follow on formula	--	59 (20.6)	10 (2.1)
Toddler Milk	--	--	64 (13.5)
Cow’s milk	--	--	165 (34.7)
Cow’s milk alternative (e.g., nut or cereal based milks)	--	--	18 (3.8)
Energy intake, mean ± SD	2553 ± 571	3009 ± 42	4247 ± 30
Under reporters *n* (%)	1 (0)	1 (0)	0 (0)
Over reporters *n* (%)	9 (10)	18 (6.3)	31 (6.5)
Energy Intake/Estimated Energy Requirement, mean (95% CI)	1.1 (1.0, 1.1)	1.1 (1.0, 1.1)	1.1 (1.1, 1.1)

^1^ Data are presented as mean ± SD, observed counts and percentages, or percentages with 95% CI for participants assigned food records; ^2^ 164/290 (57%) of young infants aged 0–5.9 months consumed only breastmilk at enrolment and were not assigned food records; ^3^ Weight-for-length z score: >+1 SD, risk of overweight; >+2 SD, overweight; >+3 SD, Obese [33]. Data were missing or implausible for *n* = 15 young infants aged 0–5.9 months, *n* = 46 infants 6–11.9 months; and *n* = 112 toddlers; ^4^
*n* = 22 young infants aged 0–5.9 m and *n* = 15 older infants aged 6–11.9 m moved into the next age bracket on the day of the food record; ^5^ Data reflects consumption on the day of the main food record for *n* = 92 young infants aged 0–5.9 months, *n* = 286 older infants aged 6–11.9 months, and *n* = 475 toddlers aged 12–24 months.

**Table 2 nutrients-14-01381-t002:** Usual energy and nutrient intake distribution from foods and beverages for mixed fed infants aged 0–5.9 months OzFITS 2021 (*n* = 90) ^1^.

	NRV Values		Distribution of Energy and Nutrient Intake						NRV Compliance (%)	
	AI ^2^	UL ^2^	10th	25th	50th	Mean ± SE	75th	90th	>AI	>UL
Macronutrients										
Energy, kJ/day	--	--	1915	2109	2477	2553 ± 60	2992	3253	--	--
Protein, g/d	10	--	9.3	10.4	12.5	12.8 ± 0.3	14.7	16.5	83	--
Protein, g/kg BW ^3^	1.43	--	1.3	1.6	2.0	2.0 ± 0.06	2.4	2.9	82	--
Fat, g/d	31	--	24	27	31	32 ± 0.8	38	42	51	--
Carbohydrates, g/d	60	--	53	58	68	69 ± 1.6	80	87	69	--
Protein, % kJ	--	--	8	8	8	9 ± 0.1	9	10	--	--
Fat, % kJ	--	--	42	45	48	47 ± 0.4	50	51	--	--
Carbohydrates, % kJ	--	--	43	45	46	46 ± 0.3	47	50	--	--
Micronutrients										
Vitamin A ^4^, µg RAE/d	250	600	303	384	512	527 ± 19.3	629	732	99	18
Thiamin, mg/d	0.2	--	0.2	0.2	0.3	0.4 ± 0.02	0.6	0.7	78	--
Riboflavin, mg/d	0.3	--	0.2	0.3	0.6	0.7 ± 0.04	1	1.3	80	--
Niacin ^5^, mg/d	2	--	4.7	6.7	10.5	10.5 ± 0.4	13.2	16.2	100	--
Vitamin B6, mg/d	0.1	--	0.1	0.2	0.3	0.3 ± 0.02	0.4	0.5	94	--
Folate, µg DFE/d	65	--	73	88	108	121 ± 4.7	150	185	92	--
Vitamin B_12_, µg/d	0.4	--	0.3	0.5	1.0	3.0 ± 1.4	2.1	4.2	82	--
Vitamin C, mg/d	25	--	23	36	54	64 ± 3.5	90	110	89	--
Calcium, mg/d	210	--	169	235	323	357 ± 18	463	560	78	--
Iron, mg/d	0.2	20	0.8	1.6	3.8	4.3 ± 0.4	6.3	8.8	99	1
Magnesium, mg/d	30	--	29	42	66	70 ± 3.4	96	113	90	--
Phosphorus, mg/d	100	--	106	137	209	225 ± 12.0	302	393	93	--
Sodium, mg/d	120	--	106	138	162	176 ± 6.1	213	248	86	--
Iodine, µg/d	90	--	70	84	102	108 ± 3.1	128	150	69	--
Selenium, µg/d	12	45	13	15	17	18 ± 0.5	22	24	92	0
Zinc, mg/d	2	4	1.8	2.3	3.6	3.8 ± 0.2	5.3	6.3	81	43

^1^ Nutrient intake data are presented as percentiles of usual intake, mean ± SE, or percentages of NRV compliance. NRV, nutrient reference values; AI, adequate intake, UL, tolerable upper level of intake; RAE, retinol activity equivalent; DFE, dietary folate equivalent; ^2^ All NRVs are from the NHMRC Nutrient Reference Values for Australia and New Zealand [4]; ^3^ BW, bodyweight; ^4^ The UL for vitamin A is for retinol only; ^5^ The AI for niacin is based on preformed niacin only.

**Table 3 nutrients-14-01381-t003:** Usual energy and nutrient intake distribution from foods and beverages for infants aged 6–11.9 months OzFITS 2021 (*n* = 286) ^1^.

	NRV Values			Distribution of Energy and Nutrient Intake						NRV Compliance (%)		
	EAR ^2^	AI ^2^	UL ^2^	10th	25th	50th	Mean ± SE	75th	90th	<EAR	>AI	>UL
Macronutrients												
Energy, kJ/day	--	--	--	2159	2565	2970	3009 ± 42	3393	3951	--	--	--
Protein, g/d	--	14	--	12	16	20	21 ± 1	26	32	--	85	--
Protein, g/kg BW ^3^	--	1.6	--	1.5	1.8	2.4	2.5 ± 0.1	2.9	3.6	--	86	--
Fat, g/d	--	30	--	24	28	33	33 ± 0.4	38	43	--	64	--
Carbohydrate, g/d	--	95	--	58	68	80	81 ± 1.2	92	107	--	23	--
Dietary fibre, g/d	--	--	--	1.9	3.7	6.0	7 ± 0.2	9	13	--	--	--
Protein, % kJ	--	--	--	9	10	12	12 ± 0.2	14	16	--	--	--
Fat, % kJ	--	--	--	35	38	42	41 ± 0.3	45	48	--	--	--
Carbohydrate, % kJ	--	--	--	41	44	46	46 ± 0.3	49	52	--	--	--
Micronutrients												
Vitamin A ^4^, µg RAE/d	--	430	600	377	468	604	626 ± 13	733	896	--	81	2
Thiamin, mg/d	--	0.3	--	0.2	0.3	0.4	0.5 ± 0.02	0.8	1.0	--	70	--
Riboflavin, mg/d	--	0.4	--	0.3	0.4	0.6	0.8 ± 0.03	1.1	1.4	--	79	--
Niacin ^5^, mg/d	--	4.0	--	11	13	15	15 ± 0.2	18	20		100	
Vitamin B6, mg/d	--	0.3	--	0.2	0.3	0.4	0.5 ± 0.01	0.6	0.7	--	72	--
Folate, µg DFE/d	--	80	--	91	120	159	179 ± 5	224	295	--	94	--
Vitamin B12, µg/d	--	0.5	--	0.5	0.7	1.1	1.4 ± 0.1	1.8	2.4	--	90	--
Vitamin C, mg/d	--	30	--	30	40	56	65 ± 2	86	114	--	90	--
Calcium, mg/d	--	270	--	185	233	329	373 ± 10	484	633	--	64	
Iodine, mg/d	--	110	--	73	85	100	100 ± 1.4	116	131	--	33	--
Iron ^6^, mg/d	7	--	20	1.1	2.1	4.3	4.9 ± 0.2	7.0	9.5	75	--	0
Magnesium, mg/d	--	75	--	48	69	99	106 ± 2.8	137	173	--	72	--
Phosphorus, mg/d	--	275	--	171	245	363	389 ± 11	513	627	--	70	--
Selenium, µg/d	--	15	60	15	18	22	23 ± 0.5	28	33	--	90	0
Sodium, mg/d	--	170	--	161	205	281	323 ± 10	400	544	--	86	--
Zinc, mg/d	2.5	--	5	2.2	2.8	3.7	4.0 ± 0.1	5.1	6.1	17	--	26

^1^ Nutrient intake data are presented as percentiles of usual intake, mean ± SE, or percentages of RDI compliance. NRV, nutrient reference values; EAR, estimated average requirement; AI, adequate intake; UL, tolerable upper level of intake; RAE, retinol activity equivalent; DFE, dietary folate equivalent; ^2^ All NRVs are from the National Health and Medical Research Council Nutrient Reference Values for Australia and New Zealand [4]; ^3^ BW, bodyweight; ^4^ The UL for vitamin A is for retinol only; ^5^ The AI for niacin is based on niacin equivalents; ^6^ The full probability approach at 10% bioavailability was used to estimate percentage at risk of inadequacy [4,35].

**Table 4 nutrients-14-01381-t004:** Usual energy and nutrient intake distribution from foods and beverages for toddlers aged 12–24 months OzFITS 2021 (*n* = 475) ^1^.

	NRV Values			Distribution of Energy and Nutrient Intake						NRV Compliance (%)		
	EAR ^2^	AI ^2^	UL ^2^	10th	25th	50th	Mean ± SE	75th	90th	<EAR	>AI	>UL
Macronutrients												
Energy, kJ/day	--	--	--	3461	3783	4233	4247 ± 30	4662	5090	--	--	--
Protein, g/d	12	--	--	30	35	39	40 ± 0.4	44	50	0	--	--
Protein, g/kg BW ^3^	0.92		--	2.6	3.0	3.5	3.6 ± 0.04	4.0	4.6	0	--	--
Fat, g/d	--	--	--	33	36	40	41 ± 0.3	44	48	--	--	--
Carbohydrate ^4^, g/d	100	--	--	92	101	115	116 ± 1.0	129	143	21	--	--
Dietary fibre, g/d	--	14	--	8.9	11.2	13.6	13.6 ± 0.2	15.7	18.3	--	43	--
Protein, % kJ	--	--	--	13	14	16	16 ± 0.1	17	19	--	--	--
Fat, % kJ	--	--	--	31	33	36	36 ± 0.2	38	40	--	--	--
Carbohydrate, % kJ	--	--	--	41	43	46	47 ± 0.2	49.5	52.2	--	--	--
Micronutrients												
Vitamin A ^5^, µg RAE/d	210	--	600	436	512	606	641 ± 8.8	736	927	0	--	0
Thiamin, mg/d	0.4	--	--	0.5	0.6	0.8	0.8 ± 0.02	1.0	1.2	3	--	--
Riboflavin, mg/d	0.4	--	--	0.7	0.9	1.2	1.2 ± 0.02	1.5	1.7	0	--	--
Niacin ^6^, mg/d	5.0	--	150	14	16	18	19 ± 0.2	21	24	0	--	0
Vitamin B6, mg/d	0.4	--	--	0.5	0.6	0.8	0.8 ± 0.01	0.9	1.2	3	--	--
Folate ^7^, µg DFE/d	120	-	300	217	260	312	320 ± 3.9	365	432	1	--	0
Vitamin B12, µg/d	0.7	--	--	1.2	1.7	2.2	2.4 ± 0.05	2.9	3.7	0	--	--
Vitamin C, mg/d	25	--	--	37	45	56	58 ± 0.9	69	83	2	--	--
Calcium, mg/d	360	--	2500	353	462	592	608 ± 9.6	743	879	11	--	0
Iodine, µg/d	65	--	200	79	94	113	116 ± 1.5	135	156	3	--	1
Iron ^8^, mg/d	4		20	3.3	4.4	5.5	5.9 ± 0.1	7.0	8.7	25	--	0
Magnesium, mg/d	65	--	65	120	151	174	179 ± 2.0	205	237	0	--	NA
Phosphorus, mg/d	380	--	3000	517	644	758	764 ± 9.2	879	1016	2	--	0
Selenium, µg/d	20	--	90	26	30	34	34 ± 0.3	38	43	0	--	0
Sodium, mg/d	--	200–400	1000	561	706	836	878 ± 12.4	1023	1210	--	100	29
Zinc, mg/d	2.5	--	7	4.2	4.7	5.2	5.3 ± 0.04	5.8	6.3	0	--	4

^1^ Nutrient intake data are presented as percentiles of usual intake, mean ± SE, or percentages of NRV compliance. NRV, nutrient reference value; AMDR, Acceptable Macronutrient Distribution Range; EAR, estimated average requirement; AI, adequate intake; UL, tolerable upper level of intake; RAE, retinol activity equivalent; DFE, dietary folate equivalent; NA, not applicable; ^2^ Unless otherwise stated, all NRVs are from the NHMRC Nutrient Reference Values for Australia and New Zealand [4]; ^3^ BW, bodyweight; ^4^ NRV is from the Institute of Medicine Dietary reference intakes for energy, carbohydrate, fiber, fat, fatty acids, cholesterol, protein, and amino acids; ^5^ The UL for vitamin A is for retinol only; ^6^ The EAR for niacin is based on niacin equivalents; ^7^ The UL for folate is for folic acid only; ^8^ The full probability approach at 15% bioavailability was used to estimate percentage at risk of inadequacy [35].

## Data Availability

The data presented in this study are available on request from the corresponding author.

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
