# Peer review of "Usual Nutrient Intake Distribution and Prevalence of Inadequacy among Australian Children 0–24 Months: Findings from the Australian Feeding Infants and Toddlers Study (OzFITS) 2021"

_nutrients, 2022, doi:10.3390/nu14071381_

Round 1

Reviewer 1 Report

This manuscript titled, “Usual Intake Distribution and Prevalence of Inadequacy among Australian Children 0-24 months: Findings from the Australian Feeding Infants and Toddlers Study (OzFITS) 2021” presents the results of a cross-sectional study to investigate usual nutrient intakes and possible inadequacies for infants/toddlers in Australia. The study is well-designed and the manuscript is well-written. The results of this study will be of interest to the readers of this journal, particularly given that there are no current data characterizing the intakes of infants and toddlers in Australia. Specific comments are listed below.  

Abstract

-See comment below in the Discussion. The evidence on sodium consumption during infancy predicting later preference for salty foods is quite limited. As a result, the authors should temper (or ideally remove) this language from the abstract.

Materials and Methods

-Second paragraph: It would be helpful to state the age range for infants eligible to participate in the study in this second paragraph.

-p. 3: The reference for the “recipe approach based on ingredients lists” for complementary foods was included, but it would be helpful to provide a few brief details of what this approach entails in the text.

Data Analysis

-It seems like the information on cases of under and over-reporting EI should be in the Results section at the end of the first paragraph. The authors state that 2 participants were excluded, but the reasons why (underreporting vs. high values) should be included in here.

Results

-Tables: The headings for macronutrients and micronutrients are in different places in each table. Consistency would be helpful.

Discussion

-The abbreviation for BMS is only used once; the authors should write out “breastmilk substitutes” again to avoid confusion.  

-“...salt may influence their preference for salty foods into adulthood” is too strong of a claim considering the available evidence on this topic. Even the review cited (#44) explains that this association “remains questionable.” As a result, the authors should temper their language both here and in the abstract.

-Page 9: The authors state that there was “no evidence of under reporting” yet in the Tables and text, the authors state that two infants had values that appeared to be under reported. Clarification is needed throughout the manuscript.  

Author Response

We thank the reviewer for their positive feedback.

Abstract

“-See comment below in the Discussion. The evidence on sodium consumption during infancy predicting later preference for salty foods is quite limited. As a result, the authors should temper (or ideally remove) this language from the abstract.”

Response: This has been amended

Materials and Methods

“Second paragraph: It would be helpful to state the age range for infants eligible to participate in the study in this second paragraph.”

Response: At Line 75 we have indicated that caregivers of infants and toddlers 0-24 months were eligible.

“-p. 3: The reference for the “recipe approach based on ingredients lists” for complementary foods was included, but it would be helpful to provide a few brief details of what this approach entails in the text.”

Response: The following has been added at Line 106

Briefly, ingredients were entered as cooked food items in FoodWorks™ based on their proportion within the ingredient list (product recipe). Different quantities were imputed for individual ingredients until a nutrient profile that closely matched the nutrition information panel was achieved. All product recipes imputed in this way were within 10% of the manufacturer reported energy, total fat, carbohydrate, protein, and total sugars. For fortified products, nutrient values generated by FoodWorks™ were replaced with manufacturer reported values.

Data Analysis

“-It seems like the information on cases of under and over-reporting EI should be in the Results section at the end of the first paragraph. The authors state that 2 participants were excluded, but the reasons why (underreporting vs. high values) should be included in here.

Response: While it is plausible that a child could consume nothing on a given day. The caregivers of these infants in the first two weeks life reported energy intakes nearly 3 times their estimated energy requirements. For example, one child was 2 weeks old had a EER of 1600 KJ but the caregiver reported an energy intake of nearly 4400 KJ.

Results

-Tables: The headings for macronutrients and micronutrients are in different places in each table. Consistency would be helpful.

Response: Thank-You this seems to have changed post-submission. We have changed it back, so the headings are now consistent

Discussion

-The abbreviation for BMS is only used once; the authors should write out “breastmilk substitutes” again to avoid confusion.  

Response: Thank you we have corrected this at Line 230

-“...salt may influence their preference for salty foods into adulthood” is too strong of a claim considering the available evidence on this topic. Even the review cited (#44) explains that this association “remains questionable.” As a result, the authors should temper their language both here and in the abstract.

Response: Agreed, at Line 262 we have tempered our language. The sentence now reads

Although the evidence base is equivocal, exposing young children to excess salt may increase their preference for salty foods later in life.

The line in the abstract has also been amended

-Page 9: The authors state that there was “no evidence of under-reporting” yet in the Tables and text, the authors state that two infants had values that appeared to be underreported. Clarification is needed throughout the manuscript.  

Response: Only two participants were classified as under-reporters out of 976 children. Thus, under-reporting is not a problem in this study.

Reviewer 2 Report

This is a well written and informative paper on nutrition in aprox 900 infants and toddlers 6 -24 months of age. The methods are clearly described and the dietary methodology is sound and designed to improve accuracy in the population.

The sample size is appropriate.

The tables are clear and easy to read and accurately represent the findings of the research.

The discussion and conclusions are reasonable and are based on the findings. I would agree with the call to have a national survey of nutrition in this population.  These findings also call into question the validity of national dietary recommendations for micronutrients based on old methodology and suggest that there may be opportunity to revisit this.

I do question why VItamin D is not discussed in the paper. It is not mentioned in the breastfeeding babies that were excluded from the study as a nutrient that may need to be supplemented, and dietary intake is not reported in the paper for the research subjects either.

I am award that Australia does not have specific vitamin D supplementation recommendations for breastfed infants in contrast to many other developed countries. Even though there are additional sources (skin / sun) for vitamin D, many parents protect their infants from the sun to avoid skin damage / cancer, and it would be an important addition to add this nutrient to the micronutrients studied in the paper.

Author Response

We thank the reviewer for their positive feedback. We agree with the reviewer about vitamin D and are also shocked that under two's have not been included in national surveys. 

"I do question why VItamin D is not discussed in the paper. It is not mentioned in the breastfeeding babies that were excluded from the study as a nutrient that may need to be supplemented, and dietary intake is not reported in the paper for the research subjects either."

"I am aware that Australia does not have specific vitamin D supplementation recommendations for breastfed infants compared to many other developed countries. Even though there are additional sources (skin/sun) for vitamin D, many parents protect their infants from the sun to avoid skin damage/cancer, and it would be an essential addition to add this nutrient to the micronutrients studied in the paper."

Response: We agree. However, Australia does not include vitamin D in its food composition databases. Mandatory fortification with vitamin D is only required for margarine, although several foods are fortified voluntarily. In our study, only a few yoghurts fortified with vitamin D were consumed. We have no data on the vitamin D status of breastfed infants in Australia outside of a few high-risk groups. The NHMRC does not perceive vitamin D as a problem. We agree that blanket vitamin D supplementation for breastfed infants is advisable, especially given the uptake of ‘SunSmart Messaging’, especially among caregivers of young children. If the reviewer wants to take on this battle, we would be supportive.

We need to be careful not to draw conclusions that are not supported by our findings.

We have added the following at Line 288.

"We did not assess the dietary intake of Vitamin D in our study due to a lack of food composition data. We recognize that the intake of vitamin D in breastfed infants is likely inadequate due to the low content in breastmilk. Unlike other high-income countries, Australian health authorities do not recommend routine vitamin D supplementation of breastfed infants."